# Relationship between Skeletal Malocclusion and Radiomorphometric Indices of the Mandible in Long Face Patients

**DOI:** 10.3390/diagnostics14050459

**Published:** 2024-02-20

**Authors:** Giuseppe D’Amato, Maryam Tofangchiha, Nima Sheikhdavoodi, Zahra Mohammadi, Mehdi Ranjbaran, Razieh Jabbarian, Romeo Patini

**Affiliations:** 1Department of Faculty of Medicine and Surgery, Unicamillus International Medical University, 00131 Rome, Italy; giuseppe.damato@unicamillus.org; 2Department of Oral and Maxillofacial Radiology, Dental Caries Prevention Research Center, Qazvin University of Medical Sciences, Qazvin 34197-59811, Iran; mt_tofangchiha@yahoo.com; 3Department of Orthodontics, School of Dentistry, Qazvin University of Medical Sciences, Qazvin 34197-59811, Iran; nima.shd@gmail.com; 4Department of Endodontics, School of Dentistry, Isfahan (Khorasgan) Branch, Islamic Azad University, Isfahan 81551-39998, Iran; 5Metabolic Diseases Research Center, Research Institute for Prevention of Non-Communicable Diseases, Qazvin University of Medical Sciences, Qazvin 34197-59811, Iran; 6Dental Caries Prevention Research Center, Qazvin University of Medical Sciences, Qazvin 34197-59811, Iran; 7Department of Pediatric Dentistry, Qazvin University of Medical Sciences, Qazvin 34197-59811, Iran; 8Department of Head, Neck and Sense Organs, School of Dentistry, Catholic University of Sacred Heart, 00135 Rome, Italy; romeo.patini@unicatt.it

**Keywords:** malocclusion, mandible, orthodontics, panoramic, radiomophometric index

## Abstract

This study aimed to assess the relationship between skeletal malocclusion and radiomorphometric indices of the mandible in long face patients. This cross-sectional study evaluated 174 lateral cephalograms and panoramic radiographs of long face patients between the ages of 17 and 30 presenting at the Orthodontics Department of Qazvin Dental School. The gonial angle, antegonial angle, type of antegonial notch, and depth of antegonial notch were measured bilaterally on panoramic radiographs. The correlation between the radiomorphometric parameters and the type of occlusion was analyzed using one-way ANOVA, independent *t*-test, Chi-square test, and Fisher’s exact test (alpha = 0.05). The mean size of gonial angle was significantly different among the three classes of occlusion (*p* = 0.046), while the difference was not significant regarding the antegonial angle size and antegonial notch depth (*p* > 0.05). An independent *t*-test showed that the mean sizes of gonial angle (*p* = 0.026) and antegonial angle (*p* = 0.036), and the antegonial notch depth (*p* = 0.046) in males, were significantly greater than the values in females. According to the Chi-square and Fisher’s exact test, the right antegonial notch type was significantly different among the three classes of malocclusion (*p* = 0.006), while this difference was not significant in the left side (*p* = 0.318). The right antegonial notch type II was more common in males, while the right antegonial notch type I was more common in females (*p* = 0.014). According to the results, the indices of gonial angle and type of antegonial notch can be clinically useful for predicting the growth rate of the mandible and designing the appropriate treatment in long face patients.

## 1. Introduction

Long face morphology is characterized by the increased vertical growth of the lower third of the face or the decreased height of the middle third of the face, and is often accompanied by incompetent lips. The situation is reverse in short face patients, and they often exhibit decreased height in the lower third of the face [1].

In general, the mandibular growth pattern plays an important role in the growth and development of the face. Finding a precise, valid, and reliable technique for mandibular growth prediction is imperative for orthodontists. Mandibular growth prediction in the early stages of life can greatly aid in more accurate diagnoses and precise treatment planning [2].

Björk A, Skieller [3] evaluated the mandibular growth pattern and showed that the majority of the vertical growth of the mandible occurs in mandibular condyles. Also, he reported that mandibles with a tendency for protrusive growth show surface apposition and surface resorption patterns below the angle of the mandible. The opposite was reported in patients whose mandibles had a tendency for retrusive growth, resulting in concavity along the lower border of the mandible, which is referred to as the antegonial notch [4,5].

The antegonial notch is located in the lower border of the mandible at the contact area of the mandibular ramus and the body of mandible, right next to the mandibular angle. It was referred to as the “pregonial notch” by MacIntosh [6], and Henderson and Poswillo [7] referred to it as the “antegonial notch”. Information regarding the presence and depth of the antegonial notch is important for maxillofacial surgical procedures conducted for the correction of mandibular disorders. Also, evidence shows that the presence of a deep antegonial notch indicates a decreased mandibular growth potential and vertical growth pattern of the mandible [8]. Clinically, it should be noted that patients with a deep antegonial notch often require longer orthodontic treatments compared with those with a shallow antegonial notch [9].

Skieller et al. [10] used multivariable statistical analyses to identify four predictive morphological variables. The detection of these variables on lateral cephalograms may predict the rotation of the mandible in the future. Although they claimed that their methodology predicted 86% of the observed changes, their study population included individuals with severely deranged morphological patterns [10].

Considering the existing controversy regarding the effect of changes in gonial and antegonial regions on mandibular growth, this study aimed to assess the relationship between skeletal malocclusion and radiomorphometric indices of the mandible (the type of antegonial notch, antegonial notch depth, and size of antegonial and gonial angles) in long face patients. The null hypothesis was that no significant correlation would be found between the skeletal class of occlusion and radiomorphometric indices of the mandible in long face patients.

## 2. Materials and Methods

This cross-sectional study was conducted on the available lateral cephalograms and panoramic radiographs of 174 long face patients presenting at the Orthodontics Department of Dental School, Qazvin, Iran, seeking orthodontic treatment from 2013 to 2020. The study was reported in accordance with the STROBE Statement, and approved by the ethics committee of Qazvin University of Medical Sciences (IR.QUMS.REC.1399.466).

The records of patients between 17 and 30 years who had presented at the Orthodontics Department of Qazvin Dental School from 2013 to 2020 seeking orthodontic treatment were evaluated. The facial height of patients was calculated on their lateral cephalograms using Jarabak cephalometric analysis. Records of patients with increased height of the lower third of the face were selected.

The exclusion criteria were previous orthodontic treatment, congenital or systemic diseases, maxillary protrusion or retrusion, decrease or loss posterior occlusal support area based on Eichner classification, cleft lip and/or palate, mandibular asymmetry and deviation, incomplete patient records, and poor quality of radiographs. Finally, 174 records were selected.

ANB angle and the Wits appraisal are the most popular cephalometric measurements applied in clinical orthodontics [10]. The skeletal classification of patients was then determined by using the Wits and Stainer analyses based on the traced lateral cephalograms by an experienced orthodontist; these were checked with the data available in patient lateral cephalograms. Lateral cephalometric was classified into Class I, Class II, and Class III according to ANB angle. Patients with ANB angle = 0 degrees, wits = −1 mm for male, and wits = 0 mm for female were categorized as class I; those with ANB > 2 degrees, wits > −1 for male, and wits > 0 for female were categorized as class II; and patients with ANB < 2 degrees, wits < −1 for male, and wits < 0 for female were categorized as class III.

Also, all the digital panoramic radiographs were traced simultaneously by an orthodontist and an oral and maxillofacial radiologist using tracing paper (Ortho Organizer); if two observers did not agree on the location of the anatomical landmarks, the third person would guide them to reach a single decision. Then, some radiomorphometric indices, including the gonial angle, antegonial angle, antegonial notch type, and antegonial notch depth, were measured by a trained general dentist (Figure 1, Figure 2, Figure 3 and Figure 4).

The depth of antegonial notch was measured on the radiographs using a digital caliper (Mitutoyo, Japan) and reported in millimeters (mm). The type of antegonial notch was determined by connecting the points A, B, and C, as shown in Figure 4 [8].

Next, the sizes of gonial angle and antegonial angle were measured and recorded. The gonial angle was measured by measuring the angle formed between the posterior and inferior borders of the mandibular ramus on the radiographs, and reported in degrees. The antegonial angle was measured by measuring the angle formed between the anterior and posterior borders of the antegonial region on the radiographs, and reported in degrees.

Data were analyzed using SPSS version 25 (IBM Corp., Armonk, NY, USA). One-way ANOVA was applied to compare the mean size of gonial and antegonial angles and the depth of antegonial notch among the three classes of occlusion. An independent *t*-test was applied to assess the correlation of the measured variables with gender. Pairwise comparisons of the classes of occlusion were carried out using the LSD post hoc test. The Chi-square test and Fisher’s exact test were applied to compare the right and left antegonial notch types based on the class of occlusion and gender. The Pearson’s correlation coefficient was applied to analyze the correlation of age with the size of gonial and antegonial angles, and the depth of antegonial notch. *p* < 0.05 was considered statistically significant.

## 3. Results

The mean age of participants was 20.95 ± 3.88 years. The participants included 110 females (63.2%) and 64 males (36.8%). Table 1 presents the frequency of patients based on the type of occlusion, and right and left antegonial notch type.

Table 2 presents the radiomorphometric indices of the mandible in long face patients. Table 3 presents the radiomorphometric indices of the mandible in long face patients based on the class of occlusion. According to one-way ANOVA, the mean size of gonial angle was significantly different among the three classes of occlusion (*p* = 0.046). The largest and the smallest size of the gonial angle were recorded in class III and class I patients, respectively. Pairwise comparisons via the LSD test indicated a significant difference between class I and class III patients in this respect (*p* = 0.019). Class I, class II, and class III patients were not significantly different regarding the antegonial angle size and antegonial notch depth (*p* > 0.05).

Table 4 compares the radiomorphometric indices of the mandible in long face patients based on gender. According to independent *t*-test, the mean sizes of the gonial and antegonial angles and the antegonial notch depth in males were significantly higher than the corresponding values in females (*p* < 0.05).

Table 5 compares the antegonial type in the right and left sides based on the class of occlusion and gender of patients using the Chi-square test and Fisher’s exact test. Significant differences were noted among the three classes of occlusion regarding the right side antegonial notch type (*p* = 0.006). The frequency of antegonial notch type I in the right side was significantly higher in class II patients and the frequency of antegonial notch type II in the right side was significantly higher in class III patients (*p* = 0.006). However, the three classes of occlusion were not significantly different regarding the antegonial notch type in the left side (*p* = 0.318).

Males and females were significantly differ ent regarding the antegonial notch type in the right side (*p* = 0.014), such that the right-side antegonial notch type II was more frequent in males, while the right-side antegonial notch type I was more frequent in females.

The Pearson’s correlation test found no significant correlation between age and size of gonial angle (r = 0.08, *p* = 0.306), age and size of antegonial angle (r = −0.01, *p* = 0.934), and age and antegonial notch depth (r = −0.06, *p* = 0.445). One-way ANOVA showed that the mean age had no significant correlation with the antegonial notch type in the right (*p* = 0.218) or left (*p* = 0.682) sides.

## 4. Discussion

This study assessed the correlation of four radiomorphometric indices of the mandible, namely the gonial angle, antegonial angle, and type and depth of antegonial notch, with the class of occlusion. The results showed that only the gonial angle and type of antegonial notch had significant correlations with the type of occlusion. A comparison of the frequency of the types of antegonial notch in the right and left sides, irrespective of the type of occlusion, revealed a significant difference between the right and left sides. The mean depth of the antegonial notch and the mean size of the gonial and antegonial angles in males were significantly greater than the corresponding values in females. The variables had no significant correlation with age.

The size of gonial angle had a significant correlation with the type of occlusion in the present study. Tayebi et al. [12] found no significant correlation between the antegonial and mental indices and type of occlusion. However, the gonial index was significantly correlated with the type of occlusion. Similar results were reported by Singer et al. [13], Lambrechets et al. [14], and the present study. However, Tayebi et al. [12] reported that the mean depth of the antegonial notch in class II patients (2.61 mm) was higher than that in class III patients (2.43 mm), while, in the present study, the mean depth of the antegonial notch in class II patients (1.77 mm) was lower than that in class III (1.80 mm) patients. This difference may be due to the fact that the lower facial height affects the antegonial notch depth. Tayebi et al. [12] did not address the lower facial height in their study. The gonial region of the mandible is the attachment site of the masseter muscle. It has been shown that the activity of the muscles of mastication, particularly the masseter, is different in patients depending on the type of occlusion [15]. Clinical studies on the role of muscles of mastication in patients with normal growth of the face have shown that muscles of mastication may apply different magnitudes of load to the sites of muscle attachment to the bone [16,17]. Thus, the increase in the thickness of the cortex of the mandible may depend on the magnitude of the applied load. Kolodziej et al. [18] longitudinally assessed 40 patients with no history of orthodontic treatment and found a significant inverse correlation between the antegonial notch depth and horizontal facial growth. They concluded that the antegonial notch depth cannot predict the future facial pattern. Similarly, Tomer and Kishnani [19] found no significant correlation between the craniofacial morphology and the antegonial notch depth. In contrast, Kar et al. [2] and Basha et al. [20] evaluated the correlation of the antegonial notch depth and facial height. They categorized the patients into three groups of normal, hyperdivergent, and hypodivergent based on the Jarabak index. They reported that the antegonial notch depth was maximum in hyperdivergent and minimum in hypodivergent patients, and this difference was statistically significant. Similar results were reported by Singer et al. [13] and Lambrechets et al. [14]. Singer et al. [13] evaluated 25 patients with deep antegonial notch who underwent orthodontic treatment for four years and showed that deep antegonial notch indicated a vertical growth pattern. Lambrechets et al. [14] evaluated 40 patients with deep and shallow antegonial notches and no history of orthodontic treatment. They concluded that patients with shallow antegonial notch mostly had a horizontal mandibular growth pattern.

The variations in the results of the available studies on this topic may be attributed to the small sample size, the long-term duration of studies, and a lack of standardization of patients regarding the horizontal growth pattern of the maxilla in the study by Kolodziej et al. [18] Moreover, Singer et al. [13] evaluated orthodontic patients, which is a confounding factor. Furthermore, it should be noted that Singer et al. [13] and Lambrechets et al. [14] evaluated patients with deep and shallow antegonial notches, indicating that their study population was not selected randomly. Also, their patients were not standardized regarding the vertical growth.

All the above-mentioned studies evaluated the correlation of antegonial notch depth and facial vertical height. The present study found no significant correlation between the antegonial notch depth and skeletal class of occlusion in long face patients. According to Salem et al. [21], due to the strategic position of the antegonial notch, this can well predict the mandibular growth pattern.

In the current study, the size of gonial angle was significantly greater in males. This finding was in agreement with the results of Jensen and Palling [22], but different from the findings of Behl et al. [23], Apaydin et al. [24], Bhardwaj et al. [25], and Mangla et al. [26]. Behl et al. [23] evaluated the radiographs of 400 Indian patients between 10 and 40 years. They found that the size of gonial angle was larger in females. Apaydin et al. [24] assessed 150 panoramic radiographs of Turkish patients between 20 and 49 years and found that the gonial angle size was larger in females. Similar results were reported by Bhardwaj et al. [25]. However, Mangla et al. [26] assessed 110 lateral cephalograms of patients between 18 and 25 years and found no significant correlation between gender and size of gonial angle. The variations in the results of studies may be due to racial and ethnic differences, different locations of the attachment of the masseter muscle at the gonion region, different patient diets, and the different masticatory forces of males and females. It should be noted that females comprised the majority of participants in the present study.

In the current study, the antegonial angle was significantly larger in males, which was in line with the results of Singh et al. [27]. However, Mangla et al. [26] found no significant correlation between the antegonial angle size and gender. The controversy in the results may be attributed to the different methods of measurement of the antegonial angle on lateral cephalograms and panoramic radiographs, as well as racial differences.

Gupta et al. [28] found a significant correlation between the antegonial notch depth and gender, and showed that the mean antegonial notch depth in male long face patients was greater than that in females, which was similar to the present findings and those of Singer et al. [13] and Kaczkowski et al. [29]; however, Mangla et al. [26] reported different results. Different antegonial notch depths in males and females could be due to the different effects of sexual hormones on bone formation and deposition [25].

In the present study, the antegonial notch type II in the right side was significantly more common in males, while the antegonial notch type I in the right side was more frequent in females. Porwolik et al. [8] reported a significant dimorphic difference in the types of antegonial notch in the left side; antegonial notch type II was more common in males, while type III was more frequent in females. The difference between their results and the present findings may be attributed to racial differences and the different measurement methods used in the two studies.

In the present study, age had no significant correlation with the measured variables. Similar results were reported by Bhardwaj et al. [25] and Chole et al. [30].

Bhardwaj et al. [25] evaluated 300 panoramic radiographs of patients in three age groups of 25–34, 35–44, and 45–54 years. They found significant correlations between the size of gonial and antegonial angles, the mandibular canal, and the mandibular foramen with gender. However, the gonial and antegonial angles had no significant correlation with age. Chole et al. [30] evaluated 1060 panoramic radiographs of patients between 15 and 66 years and found no significant correlation between age and gonial and antegonial indices. Atef et al. [31] assessed the computed tomography scans of 200 patients between 18 and 60 years in Libya and found a significant correlation between the gonial angle and the gonion–gnathion length with age. Tidke et al. [32] evaluated 200 panoramic radiographs of patients between 21 and 60 years and reported the same results. They found a significant correlation between the antegonial angle and antegonial notch depth with age. However, the findings of the aforementioned two studies were different from our results. The reason may be that the size of antegonial angle is the largest in completely dentate patients and significantly decreases in partially dentate and then in completely edentulous patients [33]. Thus, edentulism can act as a confounding factor. In the present study, the patients had a maximum of one edentulous site in each quadrant. Thus, the effect of this confounding factor on the results was eliminated, which was a strength of the present study. However, it was not among the exclusion criteria in studies by Atef et al. [31] and Tidke et al. [32].

This study had some limitations. Since the patients’ data were not available in digital form, cephalometric analysis software programs could not be used. Also, the majority of patients were females, and the age range of patients was limited. Thus, the effect of age on the variables could not be precisely evaluated. Moreover, the patients did not have 3D cone-beam computed tomography scans in their records. Future studies with longitudinal designs are required to assess the effects of puberty on radiomorphometric indices of the mandible. Also, the normal range of antegonial notch depth should be determined in future studies. The morphology of the symphysis and its effect on the facial growth pattern and type of malocclusion should also be investigated.

## 5. Conclusions

The size of gonial angle and type of antegonial notch can be clinically used to predict the growth pattern for precise treatment planning. The presence of a deep antegonial notch in patients can indicate an increased vertical growth pattern. However, the antegonial notch depth cannot predict the class of occlusion. Thus, in patients with a deep antegonial notch, measures should preferably be taken to correct facial height prior to growth spurts. However, the normal threshold for this index should first be defined.

However, it is crucial to recognize the nuanced limitations posed by the observational nature of our study, warranting careful consideration when interpreting and extending our findings. While statistical analyses have uncovered significant correlations, these findings should be viewed within the context of the inherent constraints and potential confounding elements associated with observational investigations. The need for further studies is evident, particularly to delve into the influence of age, dental status, and pubertal development on these indices, emphasizing the necessity of a broader understanding of their dynamic changes across diverse demographic groups. Moreover, efforts to determine the normative range of antegonial notch depth and explore its interplay with symphysis morphology and facial growth patterns will enrich our comprehension and guide more nuanced approaches in this domain. In essence, this study prompts a call for meticulous follow-up investigations, inviting comprehensive longitudinal explorations to unveil the complexities underlying these radiomorphometric indices and their correlations with growth patterns in long face patients.

## Figures and Tables

**Figure 1 diagnostics-14-00459-f001:**
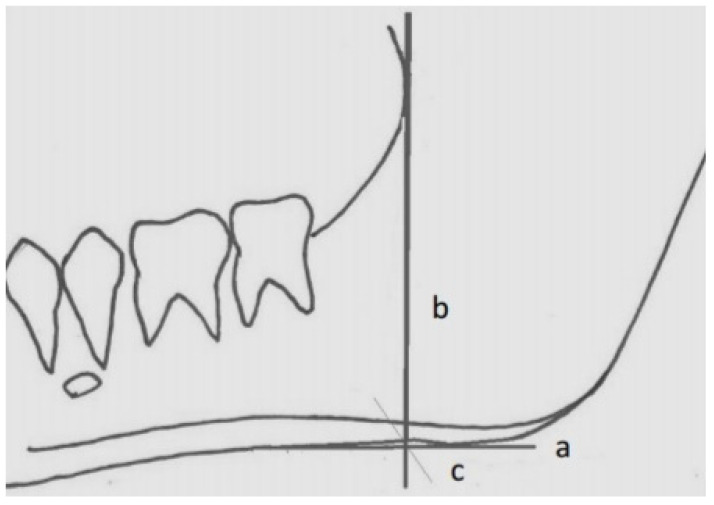
Measurements on the panoramic radiography for antegonial index. a: the inferior border of the mandible, b: the best straight line drawn along the anterior border of the ascending ramus, c: a perpendicular line is drawn to the lower cortex of the mandible from the intersection of the two lines (a and b). Antegonial Index: cortex thickness is measured in the c line.

**Figure 2 diagnostics-14-00459-f002:**
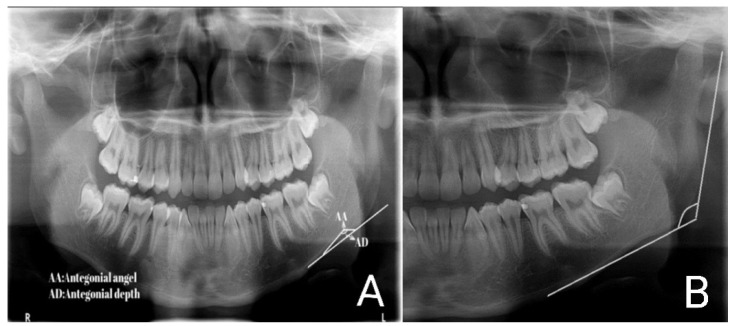
Measurements on the panoramic radiography [11]. (**A**) antegonial depth and antegonial angle, (**B**) gonial angle.

**Figure 3 diagnostics-14-00459-f003:**
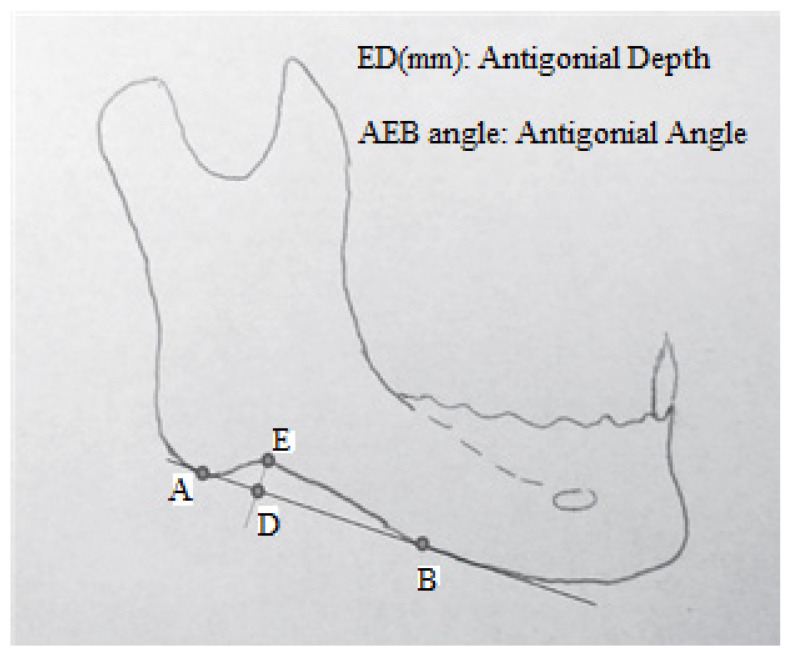
Measurements on the panoramic rediography.: antegonial depth and antegonial angle [11]. **A**: distal border of the antegonial notch. **B**: proximal border of the antegonial notch. **D**: extrapolated point of the position of the antegonial notch fundus at its basis. **E**: fundus of the antegonial notch.

**Figure 4 diagnostics-14-00459-f004:**
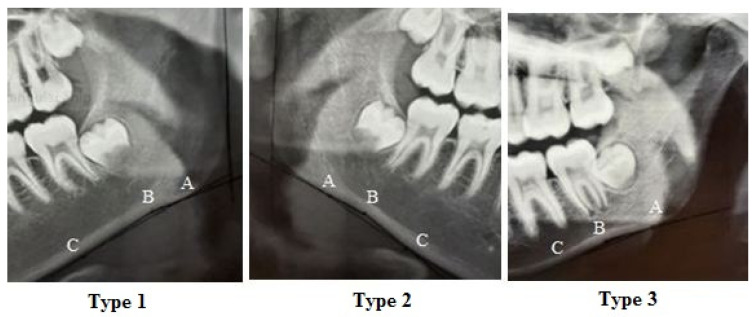
Classification of antegonial notch morphology in the panoramic radiography [11]. **A**: distal border of the antegonial notch. **B**: fundus of the antegonial notch. **C**: proximal border of the antegonial notch.

**Table 1 diagnostics-14-00459-t001:** Frequency of patients based on the type of occlusion, and right and left antegonial notch type.

Variable		Number	Percentage
Occlusion	Class I	55	31.6
	Class II	71	40.8
	Class III	48	27.6
Right antegonial notch type	Class I	92	52.9
	Class II	76	43.7
	Class III	6	3.4
Left antegonial notch type	Class I	114	65.5
	Class II	50	28.7
	Class III	10	5.7

**Table 2 diagnostics-14-00459-t002:** Radiomorphometric indices of the mandible in long face patients.

Variable	Mean	Std. Deviation	Minimum	Maximum
Right gonial angle (degrees)	126.93	7.56	107.00	146.00
Left gonial angle (degrees)	126.88	8.59	107.00	148.00
Mean right and left gonial angles	126.91	7.51	107.00	143.00
Right antegonial angle (degrees)	162.18	6.23	148.00	178.00
Left antegonial angle (degrees)	159.66	8.24	128.00	180.00
Mean right and left antegonial angles	160.92	5.79	148.50	175.00
Right antegonial notch depth (mm)	1.79	0.71	0.19	4.09
Left antegonial notch depth (mm)	1.68	0.66	0.03	3.89
Mean depth of the right and left antegonial angles	1.73	0.63	0.26	3.99

**Table 3 diagnostics-14-00459-t003:** Comparison of the mean size of gonial and antegonial angles and depth of antegonial notch in long face patients with different classes of occlusion.

Variable	Occlusion	Mean	Std. Deviation	*p*-Value *	LSD Test
Gonial angle	Class I	124.91	7.29	0.046	Class I vs. II: *p* = 0.058
	Class II	127.45	6.99		Class I vs. III: *p* = 0.019
	Class III	128.39	8.14		Class II vs. III: *p* = 0.501
Antegonial angle	Class I	162.04	5.16	0.138	-
	Class II	159.97	6.28		
	Class III	161.04	5.59		
Antegonial depth	Class I	1.63	0.56	0.313	-
	Class II	1.77	0.57		
	Class III	1.80	0.77		

* One-way ANOVA. - Since the *p*-value of the ANOVA results was not significant, therefore, LSD pairwise comparison was not performed for these variables.

**Table 4 diagnostics-14-00459-t004:** Comparison of the radiomorphometric indices of the mandible in long face patients based on gender.

Variable	Gender	Mean	Std. Deviation	*p*-Value *
Gonial angle	Male	128.68	8.60	0.026
	Female	125.87	6.62	
Antegonial angle	Male	162.02	4.34	0.036
	Female	160.28	6.42	
Antegonial depth	Male	1.86	0.62	0.046
	Female	1.66	0.62	

* Independent *t*-test.

**Table 5 diagnostics-14-00459-t005:** Comparison of the antegonial notch type in the right and left sides based on the class of occlusion and gender of patients using the Chi-square test and Fisher’s exact test.

Variable		Right Antegonial Type			Left Antegonial Type		
		Number (%)			Number (%)		
		Class I	Class II	Class III	Class I	Class II	Class III
Occlusion	Class I	32 (58.2)	21 (38.2)	2 (3.6)	34 (61.8)	19 (34.5)	2 (3.6)
	Class II	45 (63.4)	24 (33.8)	2 (2.8)	52 (73.2)	15 (21.1)	4 (5.6)
	Class III	15 (31.3)	31 (64.6)	2 (4.2)	28 (58.3)	16 (33.3)	6 (8.3)
*p*-value *		0.006			0.318		
Gender	Males	26 (40.6)	37 (57.8)	1 (1.6)	42 (65.6)	16 (25.0)	6 (9.4)
	Females	66 (60.0)	39 (35.5)	5 (4.5)	72 (65.5)	34 (30.9)	4 (3.6)
*p*-value *		0.014			0.245		

* Chi-square or Fisher’s exact test.

## Data Availability

The authors confirm that the data supporting the findings of this study are available within the article.

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
