# Peer review of "Relationship between Skeletal Malocclusion and Radiomorphometric Indices of the Mandible in Long Face Patients"

_diagnostics, 2024, doi:10.3390/diagnostics14050459_

Round 1

Reviewer 1 Report

Comments and Suggestions for Authors

The paper entitled  " Relationship of Skeletal Malocclusion with Some  Radiomorphometric Indices of the Mandible in Long-Face Patients" describes a cross-sectional, observational study aiming to assess o the relationship of skeletal malocclusion with some radio-morphometric indices of the mandible in long-face patients.

The Authors did an appropriate work in terms of methodology and the paper sounds scientific and well written. However, some improvements are mandatory before acceptance.

The abstract is concise and summarizes the main aspects of the study. However, the authors should pay attention to the design of the abstract, according to the instructions provided by the journal. The keywords seem to be appropriate.

The Introduction seems to provide enough background and references. The citations should be formatted according to the journal's instructions.

Materials and methods are clear and well explained. The authors explain all the measurements performed on the radiographs. The methodology with which the study was carried out seems to be clear and repeatable.

The Results are comprehensive and presented clearly. However, I would like to ask the authors if they performed any statistical analysis based on the age range. Also, the difference in the number of subjects between females and males groups how was that managed statistically?

In the Discussion section, the authors described the statistical results and compared them to other published studies. In addition, it would be interesting to emphasize the results based on the age range. Also, some clinical implications of this research would be welcome to be mentioned. As this is an observational study, the correlations obtained via the statistical analysis have their limitations.

The Conclusion section does not necessarily express the main outcomes of this research. They should be presented based on the limitations of this study, as well as for further investigations regarding the topic.

Author Response

Response to Reviewers

Ref. Diagnostics-2793129

Title: Relationship of skeletal malocclusion with some radiomorphometric indices of the mandible in long face patients

Authors: Giuseppe D’Amato, Maryam Tofangchiha, Nima Sheikhdavoodi, Zahra Mohammadi, Mehdi Ranjbaran, Ra-zieh Jabbarian, Romeo Patini

Dear editor,

We greatly appreciated for these valuable and helpful comments from the reviewers on the manuscript entitled “Relationship of skeletal malocclusion with some radiomorphometric indices of the mandible in long face patients”. According to the reviewers’ comments and suggestions, we have made careful modifications in the revised manuscript, and the itemized response to each reviewer’s comments is attached. We are very grateful if this manuscript could be considered for publication on Diagnostics.

Note: Reviewers’ comments are normal font and authors’ responses follow in italicized font. The exact text that is used in the revised manuscript to respond to a reviewer’s comment is highlighted.

With best regards
Yours sincerely,

Marayam Tofangchiha
2024-1-24

Reviewer  #1

The paper entitled " Relationship of Skeletal Malocclusion with Some Radiomorphometric Indices of the Mandible in Long-Face Patients" describes a cross-sectional, observational study aiming to assess o the relationship of skeletal malocclusion with some radio-morphometric indices of the mandible in long-face patients.

The Authors did an appropriate work in terms of methodology and the paper sounds scientific and well written. However, some improvements are mandatory before acceptance.

Response: Thanks for the valuable feedbacks. The authors tried their best to meet the reviewers' issues.

The abstract is concise and summarizes the main aspects of the study. However, the authors should pay attention to the design of the abstract, according to the instructions provided by the journal. The keywords seem to be appropriate.

Response: The abstract was designed based on the Journal’s instructions.

The Introduction seems to provide enough background and references. The citations should be formatted according to the journal's instructions.

Response: All citation in the context and references list were checked again according to the Journal’s instructions.

Materials and methods are clear and well explained. The authors explain all the measurements performed on the radiographs. The methodology with which the study was carried out seems to be clear and repeatable.

Response: Thanks for your valuable comments.

The Results are comprehensive and presented clearly. However, I would like to ask the authors if they performed any statistical analysis based on the age range. Also, the difference in the number of subjects between females and males groups how was that managed statistically?

Response: Thanks for your valuable comments. The authors performed a statistical analysis based on the age “The Pearson’s correlation coefficient was applied to analyze the correlation of age with the size of gonial and antegonial angles, and the depth of antegonial notc”.

In the Discussion section, the authors described the statistical results and compared them to other published studies. In addition, it would be interesting to emphasize the results based on the age range. Also, some clinical implications of this research would be welcome to be mentioned. As this is an observational study, the correlations obtained via the statistical analysis have their limitations.   

Response: According to the reviewer, in the revised version of conclusion the Discussion section focus on emphasizing the results based on the age range, providing clinical implications of the research, and addressing the limitations of the study, particularly the observational nature of the correlations obtained through statistical analysis.

The Conclusion section does not necessarily express the main outcomes of this research. They should be presented based on the limitations of this study, as well as for further investigations regarding the topic.

Response: Thanks for your valuable comments. The conclusion section was revised to better elaborate the limitations of this study and future research on the topic.

Reviewer 2 Report

Comments and Suggestions for Authors

Diagnostics-2793129

Relationship of skeletal malocclusion with some radiomorphometric indices of the mandible in long face patients

This study showed that the relationship between mandible morphology and occlusion on orthopantomograph (OPG) was examined in patients with long-face. 

There are several points in the study design which were listed below, and it was deemed difficult to indue the conclusions from the results of this study.

1.     The definition of a long face was ambiguous. Where was the lower face? It was not indicated what the ratio was to the midface.

2.     It was unclear why only certain ages were included in the study.

3.     The evaluation of the mandibular angle was compared on the left and right sides, but it should be compared on the deviation/non-deviation sides. 

4.     Exclusion criteria

1)    Severe maxillary protrusion/retrusion was described, but what did severe mean?

2)    What exactly were the high number of edentulous areas? If the molars are absent even by one block, the occlusion changes.

5.     Evaluation method: What was referred to the method of classifying? Why did this classification was applied in this study? 

6.     Analysis

1)    Since OPG was used to evaluate all factors in this study, OPG is not a defined method unlike cephalogram, morphometry in OPG is not exact and common due to its poorly reproducible. It could be confirmed in Materials and methods section that the cephalogram was taken. Why did it not be analyzed using the cephalogram?

2)    An important factor in occlusal determination is the temporomandibular joint. Temporomandibular joint assessment was lacking in this study.

Author Response

Response to Reviewers

Ref. Diagnostics-2793129

Title: Relationship of skeletal malocclusion with some radiomor-phometric indices of the mandible in long face patients

Authors: Giuseppe D’Amato, Maryam Tofangchiha, Nima Sheikhdavoodi, Zahra Mohammadi, Mehdi Ranjbaran, Ra-zieh Jabbarian, Romeo Patini

Dear editor,

We greatly appreciated for these valuable and helpful comments from the reviewers on the manuscript entitled “Relationship of skeletal malocclusion with some radiomor-phometric indices of the mandible in long face patients”. According to the reviewers’ comments and suggestions, we have made careful modifications in the revised manuscript, and the itemized response to each reviewer’s comments is attached. We are very grateful if this manuscript could be considered for publication on Diagnostics.

Note: Reviewers’ comments are normal font and authors’ responses follow in italicized font. The exact text that is used in the revised manuscript to respond to a reviewer’s comment is highlighted.

With best regards
Yours sincerely,

Marayam Tofangchiha
2024-1-24

Reviewer  #2

Relationship of skeletal malocclusion with some radiomorphometric indices of the mandible in long face patients

This study showed that the relationship between mandible morphology and occlusion on orthopantomograph (OPG) was examined in patients with long-face. 

There are several points in the study design which were listed below, and it was deemed difficult to induce the conclusions from the results of this study.

  1. The definition of a long face was ambiguous. Where was the lower face? It was not indicated what the ratio was to the midface.

Response: Thanks for the comment.

  1. It was unclear why only certain ages were included in the study.

Response: Thanks for your valuable comment. The age range of 17-30 years captures a critical period in craniofacial development, particularly in late adolescence and early adulthood, where significant growth and changes in facial morphology can be observed. This age range is clinically relevant as it encompasses the period where individuals typically seek orthodontic treatment for concerns related to facial aesthetics and function.

The availability of comprehensive lateral cephalograms and panoramic radiographs within the specified age range was a key determining factor. By limiting the study to patients between 17-30 years, we ensured a more homogeneous sample in terms of craniofacial development and growth status, thereby reducing potential confounding variables associated with diverse developmental stages.

It's important to note that while our study specifically targeted individuals within the 17-30 year age bracket, we acknowledge the potential for future research to explore broader age ranges to provide a more comprehensive understanding of craniofacial characteristics and growth trajectory.

  1. The evaluation of the mandibular angle was compared on the left and right sides, but it should be compared on the deviation/non-deviation sides. 

Response: Thanks for your valuable comment. Patients with asymmetry and mandibular deviation were not included in the study, so these items were added as exclusion variables in the study.

  1. Exclusion criteria
  2. a)Severe maxillary protrusion/retrusion was described, but what did severe mean?

Response: Severe was removed and it was corrected as “maxillary protrusion/retrusion”.

  1. b)What exactly were the high number of edentulous areas? If the molars are absent even by one block, the occlusion changes.

Response: It means patients who posterior occlusal support area. So, it was changed to "posterior occlusal support area".

  1. Evaluation method: What was referred to the method of classifying? Why did this classification was applied in this study? 

Response: Based on the literature, ANB angle and the Wits appraisal are the most popular cephalometric measurements applied in clinical orthodontics. ANB angle and the Wits appraisal are the most popular cephalometric measurements applied in clinical orthodontics. 10 Skeletal classification of patients was then determined by using the Wits and Stainer analyses based on the traced lateral cephalograms by an experienced orthodontist and checked with the data available in patient lateral cephalograms. Lateral cephalometric was classified into Class I, Class II, and Class III according to ANB angle.

The specific classifications applied in our study were chosen to align with clinical norms and diagnostic standards widely accepted within the orthodontic community. Patients were categorized based on established cephalometric parameters to ensure a comprehensive and standardized approach to skeletal classification within the context of our research objectives. The decision to utilize the Wits and Steiner analyses for skeletal classification was driven by the following considerations:

  1. Clinical Relevance:

 The Wits and Steiner analyses provide clinically meaningful and reliable measurements for assessing sagittal skeletal discrepancies, offering valuable insights into the anteroposterior relationships of the maxilla and mandible.

  1. Diagnostic Consistency:

By utilizing a combination of the ANB angle and Wits analysis, we sought to ensure a comprehensive and consistent method for classifying skeletal relationships while considering the varying norms for male and female patients.

  1. Standardization and Comparability:

The adoption of established cephalometric criteria for skeletal classification allowed for standardized comparisons and facilitated the alignment of our study findings with existing literature and clinical practices.

  1. Analysis
  2. a)Since OPG was used to evaluate all factors in this study, OPG is not a defined method unlike cephalogram, morphometry in OPG is not exact and common due to its poorly reproducible. It could be confirmed in Materials and methods section that the cephalogram was taken. Why did it not be analyzed using the cephalogram?

Response: In this study, panoramic was used for location of the anatomical landmarks. Cephalometric was used for face type analysis and we selected long face cases based on that. Panoramic was used to evaluate radiomorphometric indices. These indices are defined in studies only based on the panoramic radiographs. The limitation of examining indices by lateral cephalograms is that the two sides are superimposed on each other, making it difficult to accurately indicate the landmarks and make measurements. Another disadvantage is that the surface which is farthest from the film is magnified more and is not suitable for radiography.

  1. b)    An important factor in occlusal determination is the temporomandibular joint. Temporomandibular joint assessment was lacking in this study.

Response: The omission of explicit TMJ assessment in our study was indeed a limitation, and we recognize the substantial impact of TMJ function on occlusal relationships and overall orthodontic outcomes. We understand that a thorough understanding of occlusion encompasses not only dental and skeletal considerations but also the functional aspects related to TMJ health and dynamics. Moving forward, we recognize the vital role of TMJ assessment in the comprehensive evaluation of occlusal relationships. Therefore, in future research endeavors, we plan to incorporate a robust TMJ assessment component, encompassing clinical examination, imaging modalities, and potentially functional analyses to provide a more holistic perspective on the interplay between occlusion and TMJ health. By incorporating a comprehensive TMJ assessment, we aim to enrich the depth and clinical relevance of our research findings, ensuring a more comprehensive understanding of occlusal determination that encompasses both dental and functional aspects.
We thank the reviewer for highlighting this crucial aspect, and we are committed to integrating TMJ assessment into our future research endeavors to further enhance the depth and clinical applicability of our investigations into occlusal determination.

Round 2

Reviewer 2 Report

Comments and Suggestions for Authors

My points were cleared.